ecology/ecosystems/medicinal chemistry

DPPH, alkaloids, terpenoids, flavonoids, phenols

**Author for correspondence:**
H. Michael G. Lattorff
e-mail: mlattorff@icipe.org

†Present address: University of Nairobi, PO Box 30197-00100, Nairobi, Kenya.

This article has been edited by the Royal Society of Chemistry, including the commissioning, peer review process and editorial aspects up to the point of acceptance.

# Phytochemical composition and bio-functional properties of *Apis mellifera* propolis from Kenya

Timothy M. Kegode[1,2], Joel L. Bargul[1,2],
Hosea O. Mokaya[1] and H. Michael G. Lattorff[1,†]

[1]International Centre of Insect Physiology and Ecology (icipe), PO Box 30772-00100, Nairobi, Kenya
[2]Biochemistry Department, Jomo Kenyatta University of Agriculture and Technology, PO Box 62000-00200, Nairobi, Kenya

TMK, 0000-0002-6888-5541; JLB, 0000-0001-8573-6807;
HOM, 0000-0003-4779-5309; HMGL, 0000-0002-8603-6332

There is an increased demand for natural products like propolis, yet little information is available about the chemical composition of African propolis and its bio-functional properties. Therefore, in this study, we aimed to quantify the phytochemicals and determine the antioxidant and antimicrobial properties of *Apis mellifera* propolis ($n = 59$) sourced from various regions in Kenya. Principal component analysis (PCA) showed that the sampling region had a remarkable impact on the propolis's composition and bio-functional properties. Generally, the propolis contained high amounts of phytochemicals, particularly alkaloids (5.76 g CE/100 g) and phenols (2.24 g GAE/100 g). Furthermore, analysis of propolis by gas chromatography–mass spectrometry (GC-MS) revealed various compounds with varying bio-functional activities. These compounds included triterpenoids alpha- and beta-amyrin, oleanen-3-yl-acetate, urs-12-en-24-oic acid, lanosta-8,24-dien-3-one, and hydrocarbons tricosane and nondecane, which have been reported to have either antimicrobial or antioxidant activities. The propolis samples collected from hotter climatic conditions contained a higher composition of phytochemicals, and additionally, they displayed higher antioxidant and antimicrobial activities than those obtained from cooler climatic conditions. Key findings of this study demonstrate the occurrence of relatively high phytochemical content in Kenya's propolis, which has antioxidant and antimicrobial properties; hence this potential could be harnessed for disease control.

**ROYAL SOCIETY OF CHEMISTRY**

# 1. Introduction

Propolis is a natural resinous mixture comprising plant resins, bee saliva and wax, which is used by wild and feral bees as a building material for sealing unwanted spaces in the hive, hence smoothening it [1]. In addition, propolis plays a crucial role in chemical defence against invading microorganisms, thus making the hive environment sterile [2]. Furthermore, bees use propolis to embalm dead and live intruders such as small hive beetles [3] and other organisms that are too large to be removed [4,5].

Further, the beneficial therapeutic effects of propolis have been recognized and applied since ancient times. Historical records suggest that its use dates back to ancient Greeks and Romans [6]. The bioactive properties of propolis, such as anti-inflammatory, anti-cancer, antioxidant, antimicrobial and immune-stimulating activities, have all been demonstrated by previous studies [7,8]. The chemicals responsible for these bioactive properties have been shown to include flavonoids and phenols, among others [9]. In recent years, modern societies have recognized and appreciated natural propolis-based treatments for disease control and management and are therefore embracing the development of these products [10].

Currently, lifestyle diseases, including but not limited to arthritis, cardiovascular disease, cancer, diabetes, Parkinson's and Alzheimer's disease, are affecting an increasing number of people [11]. This is attributed to free radicals, chemical and physical factors leading to cell ageing, the genesis of pathophysiological processes [12]. Propolis is a source of natural antioxidants, scavenging and acting as secondary defensive factors against free radicals produced in the human body [13]. Enzymes serve as the primary defence system against oxidative stress as a result of normal body physiological processes. These enzymes include catalase, glutathione peroxidase and superoxide dismutase [14]. When the generation of free radicals overcomes the primary buffer capacity of the human body, the second line of defence comprises vitamins [15]. These vitamins, for example vitamins C and E, play roles by scavenging free radicals and inactivating them [10]. Exhaustion of these two defence mechanisms leads to severe cell damage in the body [16]. Phenolic compounds, flavonols, flavones, flavanones and isoflavanones are important bioactive constituents of propolis with a proven ability to scavenge free radicals, apart from shielding vitamin C and lipids denaturation by the resulting oxidative physiological process [17].

Disease-causing microbes are evolving to adapt to the constant exposure to antimicrobial agents leading to the emergence of multidrug-resistant pathogens [18]. This has made the treatment of diseases difficult and costly. Methicillin-resistant *Staphylococcus aureus* (MRSA) is an example that could cause disfiguring of the affected body part or loss of life if left untreated [19–21]. These challenges have inspired research on better and more efficient alternatives. This led to a re-evaluation of the therapeutic properties of natural products exhibiting antimicrobial activity, such as propolis [22]. The potency of propolis as a natural bioactive agent against both Gram-positive and Gram-negative microorganisms like multidrug-resistant bacteria was demonstrated by [1], indicating its efficacy. Turkish propolis was also efficacious against tuberculosis, and the results pointed out its potential activity against different species of mycobacteria [23]. These properties result from diverse antimicrobial factors such as flavonoids, phenols and other compounds [24]. The clinical antibacterial properties of propolis have been supported due to the myriad bioactive compounds present in it [25]. Therefore, propolis has attracted much interest in its application to treat various maladies. Consequently, propolis has been used to develop new drugs through biotechnology [25].

Presently, it is unclear how the vegetation serving as the source of resins in the propolis influences the phytochemical composition and the biological activity of propolis. Often, the plants used for resin collection by bees are unknown. Thus, we can use environmental variables, such as climatic variables, vegetation and land use/land cover (LULC) parameters as proxies to predict propolis's phytochemical composition and biological activity. In this study, we analysed the phytochemicals and determined the antioxidant and antibacterial activities of *Apis mellifera* propolis collected from various geographically distinct regions in Kenya, which differ in climatic and vegetation parameters. The results may help understand how propolis' properties vary according to regions and climatic conditions that influence it.

# 2. Material and methods

## 2.1. Chemicals

Mueller–Hinton Agar (MHA) was purchased from Himedia Laboratories Pvt. Ltd (F&S Scientific, Nairobi, Kenya). Sodium nitrite (NaNO$_2$), 1,10-phenanthroline, aluminium chloride (AlCl$_3$), Gallic acid, linalool, quercetin, sodium carbonate (Na$_2$CO$_3$), Folin–Ciocalteu's reagent, sodium hydroxide (NaOH),

**Table 1.** Sampling regions, locations and their different climatic conditions. Sample sizes with respect to apiaries and individual propolis samples are given in the last two columns.

| region | location | climatic condition | apiaries (N) | propolis samples (N) |
|---|---|---|---|---|
| Rift Valley | Marigat | hot, dry | 3 | 9 |
| Central | Murang'a | cold, wet | 6 | 8 |
| Eastern | South Kitui | hot, dry | 5 | 5 |
| Western | Kakamega | hot, wet | 5 | 10 |
| Nairobi | Karura | hot, dry | 3 | 6 |
| | icipe | cold, wet | 1 | 3 |
| Coast | Gede | hot, wet | 3 | 4 |
| | Mtwapa | hot, wet | 4 | 6 |
| | lower Taita | hot, dry | 1 | 3 |
| | Taita Hills | cold, wet | 2 | 5 |

hydrochloric acid (HCl), chloroform, dichloromethane (DCM), sulfuric acid ($H_2SO_4$), colchicine, ferric-III-chloride ($FeCl_3$), absolute ethanol and 2,2-diphenyl-1-picrylhydrazyl (DPPH) were all purchased from Sigma-Aldrich (Kobian Kenya Ltd). For all analyses, we used chemicals of analytical grade.

## 2.2. Collection of propolis

Fifty-nine propolis samples were collected directly from hives by scraping with a hive tool and wrapping them with aluminium foil. Sampling was done by randomly selecting two hives per apiary from each location in the six different regions in Kenya with differing climatic conditions (table 1). The propolis samples were stored at −80°C, awaiting further processing.

## 2.3. Sample preparation

The individual propolis samples were crushed into a fine powder in liquid nitrogen using a mortar and pestle. Extracts for the analysis of the phytochemical composition and radical scavenging activity were prepared by weighing 0.5 g of each crushed sample in 10 ml of 50% v/v ethanol in 15 ml Falcon tubes and left to stand at room temperature for 72 h. They were vortexed at 3000 r.p.m. for 3 min, followed by the second vortexing step after 24 h of incubation at room temperature. After the extraction, the samples were centrifuged at 600 r.p.m. for 2 min, and the supernatant was decanted and stored at −80°C. For testing the antibacterial activity, the extractions were conducted by dissolving 2 g of the propolis sample in 10 ml of absolute ethanol (100%). The mixture was vortexed at 3000 r.p.m. for 2 min and then incubated for 24 h at room temperature. The subsequent extraction followed the same steps described above, and concentration was done in an Eppendorf concentrator plus (Eppendorf, Hamburg, Germany) to complete dryness. The residue was weighed on an analytical plus scale (OHAUS Corporation Parsippany, New Jersey, USA) and dissolved in absolute ethanol to make a concentration of 80 mg ml$^{-1}$.

## 2.4. Quantification of phytochemicals

### 2.4.1. Flavonoid content

The total flavonoid content (TFC) of propolis samples was done following the aluminium chloride ($AlCl_3$) colorimetric assay described by Popova et al. [26]. To 1 ml extract of each sample, 4 ml of distilled water was added, followed by 300 µl of 5% (w/v) $NaNO_2$ and mixed. After 5 min, 300 µl of 10% $AlCl_3$ was added to the mixture and left for 1 min prior to adding 2 ml of 1 M NaOH, followed by a top-up step with 2.4 ml of distilled water. The absorbance of the mixture was measured at 510 nm using a spectrophotometer (BioSpec, Bartlesville, USA) against a reagent blank containing all the above reagents, except the sample that was replaced with 50% ethanol. Quercetin at different concentrations (20–200 µg ml$^{-1}$) was used as standard to generate a calibration curve ($y = 0.0006x + 0.0028$, $R^2 = 0.9981$) and TFC expressed as mg quercetin equivalent per 100 g propolis (mg QE/100 g propolis).

### 2.4.2. Phenol content

We used the Folin–Ciocalteu method described by Popova et al. [26] to determine the total phenol content (TPC). To 1 ml extract, 5 ml of 0.2 N Folin–Ciocalteu reagent was added and left at room temperature for 5 min. After adding 4 ml of 75 g l$^{-1}$ Na$_2$CO$_3$, the mixture was incubated at room temperature for 2 h. The absorbance of this reaction mixture was read at 760 nm against an ethanol blank instead of methanol. Gallic acid at different concentrations (0–250 µg ml$^{-1}$) was used as a standard to yield a calibration curve ($y = 0.0073x + 0.0233$, $R^2 = 0.9992$). All the assays were done in triplicates, and the mean obtained was used to calculate the total phenol content, TPC, in propolis samples. The TPC was expressed as mg gallic acid equivalents (mg GAE/100 g propolis).

### 2.4.3. Alkaloid content

The propolis samples' total alkaloid content (TAC) was measured using the 1,10-phenanthroline method as previously described by Pandey et al. [27]. One millilitre of 0.025 M FeCl$_3$ in 0.5 M HCl was mixed with 1 ml propolis extract, followed by the addition of 1 ml of 0.05 M 1,10-phenanthroline in 50% (w/v) ethanol. The mixture was incubated for 30 min in a water bath at 70°C. The absorbance of the red-coloured complex was measured at 510 nm against a reagent blank containing the reagents only without the sample. Total alkaloid content was estimated from the standard curve plotted using 0.1–1.5 mg ml$^{-1}$ colchicine ($y = 1.866x + 0.2332$, $R^2 = 0.9844$). Thus, 10 mg of colchicine was dissolved in 10 ml of 50% ethanol (w/v) to generate seven data points conducted in triplicates to allow computation of the means for plotting the standard curve. The TAC was expressed as mg colchicine/100 g propolis.

### 2.4.4. Terpenoid content

The total terpenoid content (TTC) was quantified using the colorimetric method described by Malik et al. [28]. First, 1.5 ml chloroform was added to a 200 µl propolis sample, and the mixture was vortexed at 3000 r.p.m. for 3 min. Then, 100 µl of concentrated H$_2$SO$_4$ was added to the mixture and incubated in the dark for 90 min. The supernatant was decanted gently to leave the reddish-brown precipitate forming at the bottom. Next, 1.5 ml of absolute methanol was added to the precipitate and the resulting mixture vortexed again to dissolve the precipitate completely. The same procedure was repeated for the linalool standard using different concentrations of linalool (10–500 mg ml$^{-1}$). The absorbance was measured at 538 nm with methanol as the blank. TTC was calculated using the linalool standard curve ($y = 0.0009x - 0.0158$, $R^2 = 0.9914$) and expressed as linalool equivalents in 100 g propolis (mg LE/100 g propolis).

## 2.5. Antimicrobial activity

### 2.5.1. Bacterial growth and maintenance

Single bacterial colonies of Gram-negative Escherichia coli from an overnight culture on Mueller–Hinton Agar (MHA) were inoculated into sterile distilled water to achieve turbidity of 0.5 McFarland ($\approx 1 \times 10^8$ CFU ml$^{-1}$ as per Clinical and Laboratory Standards Institute) by measuring the optical density (OD) of 0.132 at 600 nm. The same procedure was repeated for the Gram-positive Bacillus thuringiensis.

### 2.5.2. Disc diffusion assay (Kirby-Bauer test)

This assay was performed in sterile MHA prepared in separate sterile Petri dishes (diameter 90 mm). From an overnight culture of E. coli prepared as mentioned above, 25 µl were spread on each agar plate using 10 silica beads for homogeneous spreading. Four circular filter paper discs (Whatman, Maidstone, UK) (8 mm diameter) were cut using a sterilized metallic borer and placed on the surface of the agar plates containing the bacteria. To the discs, 25 µl of 80 mg l$^{-1}$ propolis sample extracts dissolved in absolute ethanol were introduced. The plates were incubated for 24 h at 37°C alongside a negative control using the solvent (ethanol) instead of propolis and a positive control containing streptomycin at the same concentration of 80 mg l$^{-1}$ and volume of 25 µl as used for the samples. Digital pictures of Petri dishes with zones of inhibition were recorded using a digital camera, and the zone diameters were measured using ImageJ software [29]. Each sample was assayed in triplicate. The above procedure was repeated for B. thuringiensis.

## 2.6. Antioxidant activity

### 2.6.1. Analysis of *in vitro* DPPH (2,2-diphenyl-1-picrylhydrazyl) radical scavenging activity

The DPPH assay was performed as outlined by Lagouri *et al.* [12] with minor modifications. Briefly, to 1.5 ml of the propolis sample solution, 3 ml of DPPH ethanolic solution (2 mg/100 ml ethanol) were added. The mixture was incubated for 45 min at 37°C in the dark, and the absorbance was measured at 517 nm. Methanol was used as control instead of propolis. For positive control, quercetin at different concentrations (10–100 µg ml$^{-1}$) was used to prepare a standard curve ($y = 0.591x + 38.413$, $R^2 = 0.9988$) and the results tabulated as quercetin equivalent. Each sample was assayed in triplicate, and the results were averaged and used to calculate the antioxidant activity as free radical scavenging activity expressed as a percentage of inhibition, using the following formula:

$$\%\text{inhibition} = [(\text{control absorbance}-\text{sample absorbance})/\text{control absorbance}] \times 100$$

## 2.7. Gas chromatography–mass spectrometry analysis

### 2.7.1. Liquid–liquid extraction and analysis

Based on the phytochemical contents and level of bioactivity, representative samples from hot and dry climatic conditions and cooler climatic conditions were chosen for GC-MS analysis. This was done using a gas chromatograph (HP-7890A, Agilent Technologies, USA) coupled with a mass spectrometer (MS-597C, Agilent Technologies, USA). The samples were extracted using absolute ethanol, and the supernatant evaporated to complete dryness leaving a solid residue. The residue was redissolved in dichloromethane to make one part per million and subjected to the GC-MS analysis. Chromatographic separations were achieved by a HP-5MS capillary column, 30 × 0.25 mm i.d., 0.25 µm thick (J & W Scientific, USA) immobilized with 5% (phenylmethyl silicone) as the stationary phase.

In the splitless mode, 1 µl of the propolis sample was injected into the GC-MS instrument using an autosampler (7683B series, Agilent Technologies, USA). The sample was then transported by helium (99.99% purity) as the carrier gas at a flow rate of 1.2 ml min$^{-1}$. First, the oven temperature was programmed at 35°C, where it was held for 5 min, followed by a gradual increase at the rate of 10°C min$^{-1}$ to 280°C, where it was held at an isothermal state for 30 min.

## 2.8. Data analysis

R working environment v. 3.5.0 (R Core Team 2019) incl. packages *factoextra 1.0.5* and *ggplot2 3.1.1* were used to perform a principal components analysis (PCA) to analyse and visualize the phytochemical composition of propolis from the different geographical regions. A Kruskal–Wallis test was used to compare phytochemical contents and the biological activities of propolis samples from different locations at $p < 0.05$, with Dunn's test for pair-wise comparison between the regions. A Spearman's rank correlation was performed to evaluate relationships between the studied parameters.

Extracted chemical components were identified by name and area under the peak as a proxy for the quantity by comparing the retention time of the chromatographic peak with the WILEY8 database combined with NIST library v. 2.0. The data presented had a similarity structure compound estimate (similarity index) greater than or equal to 90%. Prediction of biological activity of the compounds is based on Dr Duke's Phytochemical and Ethnobotanical Databases created by Dr Jim Duke of Agricultural Research Service/USDA-ARS in 2016.

# 3. Results

## 3.1. Quantification of phytochemicals

The four quantified phytochemical groups showed significant variation among all the locations based on the colorimetric assays (table 2) with $p \leq 0.05$. The total content of individual phytochemicals varied in each sample, with alkaloids being most abundant, followed by phenols, flavonoids and terpenoids, respectively. The mean total quantity of phenol content was between 522.6 and 3,711.8 mg GAE/100 g

**Table 2.** Means ± standard deviation of the four phytochemicals in propolis showing variations between the regions and individual locations (mg/100 g of propolis). Different superscript letters in each column show variations in samples collected from different locations as revealed by the pairwise comparisons (Dunn's test). QE = Quercetin Equivalent, GAE = Gallic Acid Equivalent, LE = Linalool Equivalent, CE = Colchicine Equivalent.

| region | location | flavanoids (mg QE/100 g) | phenols (mg GAE/100 g) | terpenoids (mg LE/100 g) | alkaloids (mg CE/100 g) |
|---|---|---|---|---|---|
| Rift Valley | Marigat | 2131.4 ± 1132.4[c] | 2732.7 ± 672.1[c,d] | 471.1 ± 270.5[b] | 4651.5 ± 138.3[a,b,c] |
| Central | Murang'a | 412.2 ± 276.0[a] | 1712.4 ± 496.6[b,c] | 272.0 ± 90.0[a] | 4672.2 ± 181.7[a,b,c] |
| Eastern | South Kitui | 1388.9 ± 1065.6[b,c] | 3161.2 ± 2339.1[c,d] | 475.7 ± 93.0[b] | 7494.7 ± 4100.1[c] |
| Western | Kakamega | 1153.3 ± 488.1[b,c] | 853.0 ± 506.3[a,b] | 347.7 ± 61.5[b] | 4196.4 ± 426.6[a] |
| Nairobi | icipe | 1717.8 ± 48.4[b,c] | 3711.8 ± 429.7[d] | 202.8 ± 59.5[a] | 4453.1 ± 98.4[a,b] |
| | Karura | 1112.2 ± 143.6[b,c] | 2218.5 ± 380.1[c,d] | 301.4 ± 57.3[a] | 4435.7 ± 131.1[a] |
| Coast | Gede | 456.7 ± 124.9[a,b] | 1225.4 ± 190.4[a,b,c] | 407.3 ± 134.6[b] | 7698.7 ± 835.1[d] |
| | lower Taita | 825.2 ± 361.1[a,b,c] | 2813.8 ± 323.3[c,d] | 353.1 ± 22.9[b] | 8767.5 ± 257.4[d] |
| | Mtwapa | 1391.1 ± 658.1[b,c] | 1726.1 ± 1283.9[a,b,c] | 319.4 ± 104.5[a] | 7614.7 ± 2807.2[b,c] |
| | Taita Hills | 102.2 ± 68.3[a] | 522.6 ± 169.3[a] | 258.5 ± 21.40[a] | 3014 ± 2692.7[a] |
| | p-value | 0.0002 | 0.00006 | 0.0005 | 0.002 |

of propolis, flavonoid content ranged from 102.2 and 3324.4 mg QE/100 g, alkaloid content from 2013.9 to 8767.5 mg CE/100 g and terpenoid content from 202.8 to 582.1 mg LE/100 g.

A pairwise comparison by Dunn's test showed how samples differed from each other. The locations had different climatic conditions (table 1), and those with hot and dry climatic conditions exhibited high total phytochemical content compared with the other locations. The terpenoid content indicates the clear difference from the locations, categorizing them only into two groups.

## 3.2. Antioxidant activity

The samples had an antioxidant activity based on their ability to scavenge DPPH free radicals leading to a colour change from purple to colourless. There is a difference in antioxidant activity of propolis ranging from 15 to 74% RSA for 250 mg l$^{-1}$ of propolis extract (figure 1). The quercetin equivalent from the standard curve indicates the average inhibition at 0.25 µg ml$^{-1}$ (250 mg l$^{-1}$) to be equivalent to 23.0 µg ml$^{-1}$ of quercetin (electronic supplementary material, table S2). The samples with lower equivalence are a result of the very low concentration of our samples, which means at the same concentration as the standard they will have a stronger inhibition. There is a variation in regions, with icipe, Marigat and Murang'a having all samples with high antioxidant activity, while Kitui and Mtwapa had samples with considerable variations, with some having the highest levels compared with others. Karura and Kakamega had samples with almost the same antioxidant activities, with the same pattern exhibited by Mtwapa and Gede. Taita had the lowest antioxidant activity.

## 3.3. Antimicrobial activity

All the tested propolis samples exhibited broad-spectrum antibacterial activity. They inhibited the growth of *E. coli* and *B. thuringiensis*, which are Gram-negative and Gram-positive bacteria, respectively (figure 2). Compared with the positive control, *E. coli* had an average inhibition of 32.5% and *B. thuringiensis* 39.4% (electronic supplementary material, table S4). Propolis has elevated activity against *B. thuringiensis* compared with *E. coli*. Propolis from Gede exhibited the highest inhibition against Gram-positive bacteria, while Kitui, Mtwapa, lower Taita and Marigat had the highest against Gram-negative bacteria.

From the PCA plot (figure 3), it can be inferred that propolis' phytochemicals are separated according to locations. Major separation is on Dimension (Dim) 1, which separates all the regions into two, with Kitui, Gede, *icipe*, Mtwapa, lower Taita and Marigat on the negative side while Karura, Murang'a, Kakamega and Taita Hills are on the positive side. The regions on the negative side experience hot climates while the others have cold climates. Dim 2 separates the components based on phytochemical content, with flavonoids and terpenoids being on the positive side while alkaloids and phenols are on the negative side.

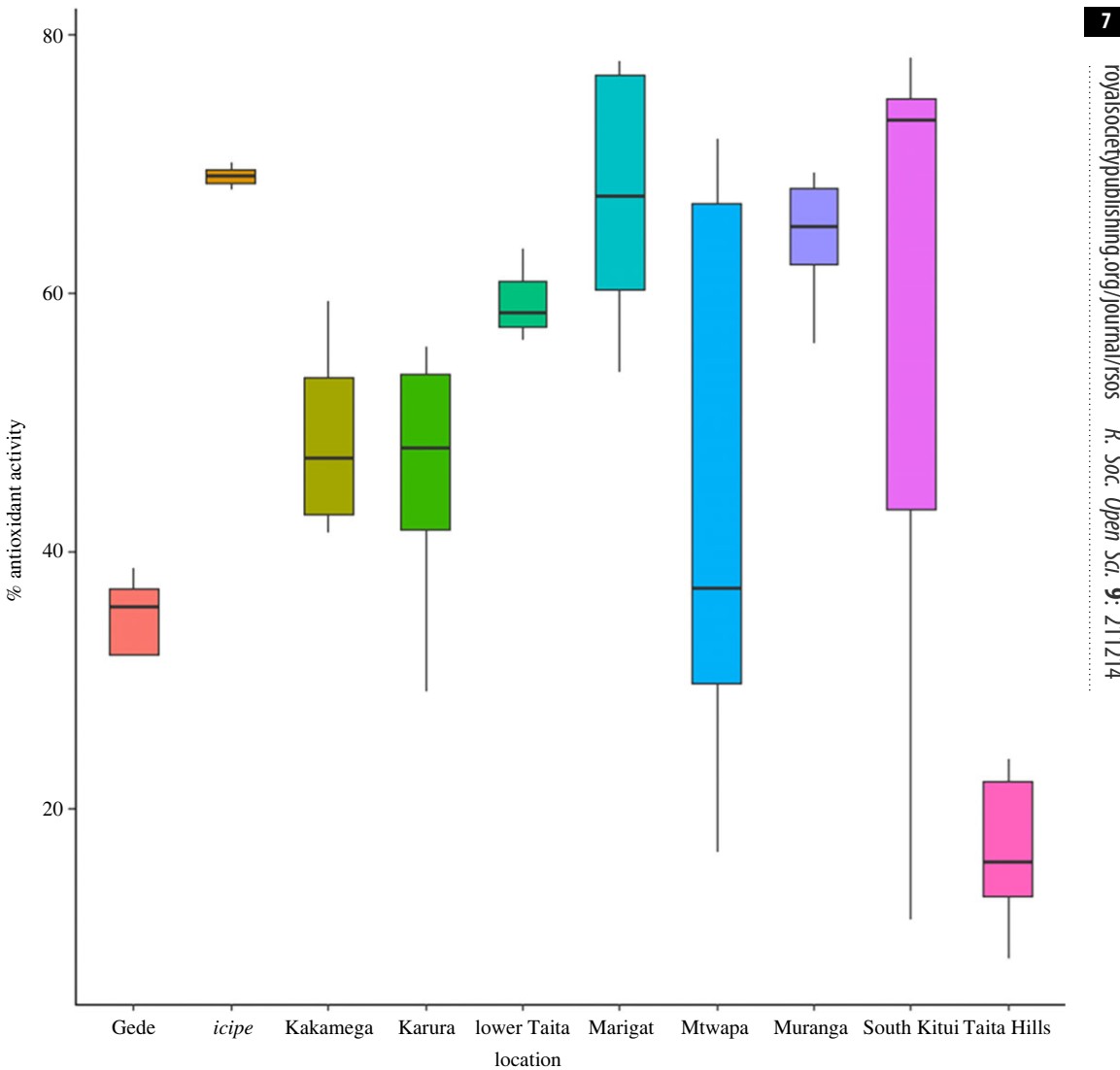

**Figure 1.** Antioxidant activity of Kenyan propolis samples according to their respective sampling locations.

From the Spearman rank correlation plot (figure 4), we inferred a significant correlation between phytochemicals and the biological activities of propolis with phenols and antioxidant activity ($r = 0.78$), and alkaloids and antibacterial activity against *E. coli* ($r = 0.61$) showing strong correlations.

## 3.4. Gas chromatography–mass spectrometry analysis

Analysis of GC-MS chromatograms of the selected propolis samples showed more than 100 peaks indicating the presence of various compounds. These compounds belong to different classes: hydrocarbons, terpenoids, diterpenoids, triterpenoids and phenols (table 3). Terpenoids are the most abundant substances as they are volatile organic compounds and hence best to detect in GC-MS analysis. Hydrocarbons also are volatile but most in propolis are long chained; hence they were also detected in moderate abundance. The flavonoid and phenol classes of compounds were the least abundant. There was a clear difference in the compounds present in all the locations, with those from hot and dry climates having more compounds than the other regions. The climatic conditions contribute to the diverse chemical composition, with only two compounds being present in all the samples, octadecahydro-2H-picen-3-one and eicosane, but with varying abundance. All the compounds reported have biological activities except nonadecane, a hydrocarbon. These compounds result in propolis's antioxidant and antibacterial activities (figure 5) because their separation and indication of specific compounds have a more pronounced effect on either antioxidant activity or antibacterial activity. The antibacterial activity is a result of compounds that exhibit broad-spectrum antibacterial activity.

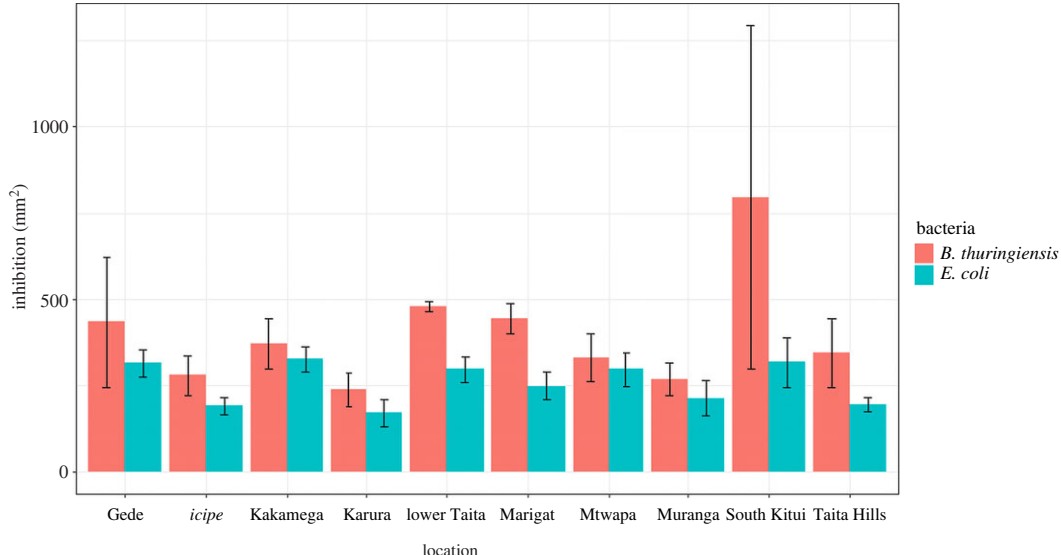

**Figure 2.** Antibacterial activity of Kenyan propolis samples for the two tested bacteria, *E. coli* and *B. thuringiensis*.

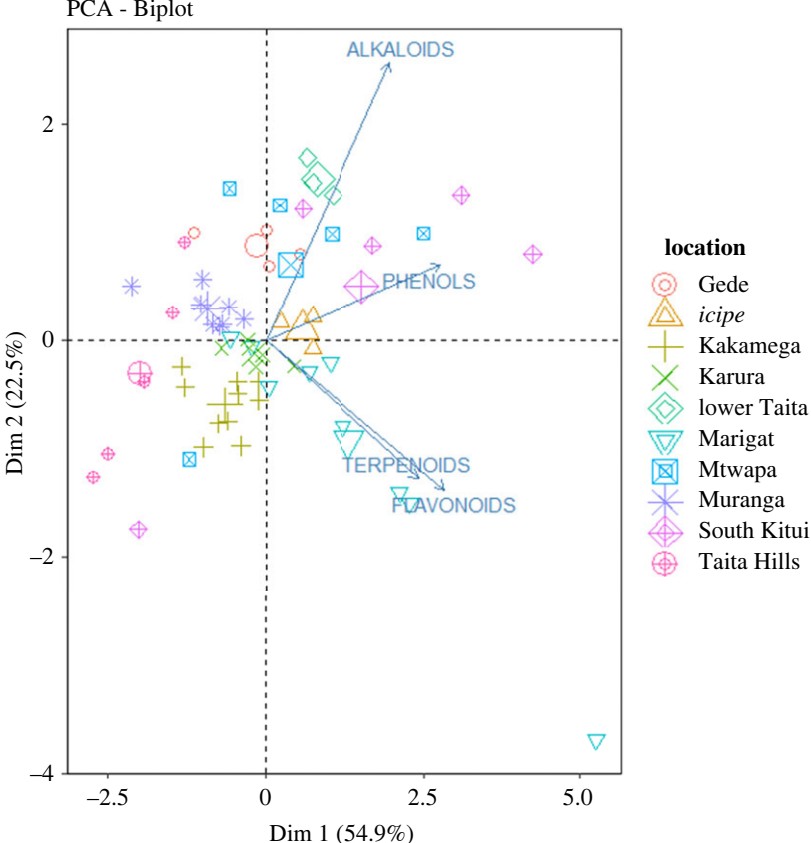

**Figure 3.** Principal components analysis of Kenyan propolis samples based on their phytochemical content showing clear separation among the sampling locations. Total variance explained: 77.4%.

# 4. Discussion

## 4.1. Quantification of phytochemicals

Varying amounts of phytochemicals illustrate the complex chemical composition of propolis. The concentration of phytochemicals is dependent on the geographical origin and its associated

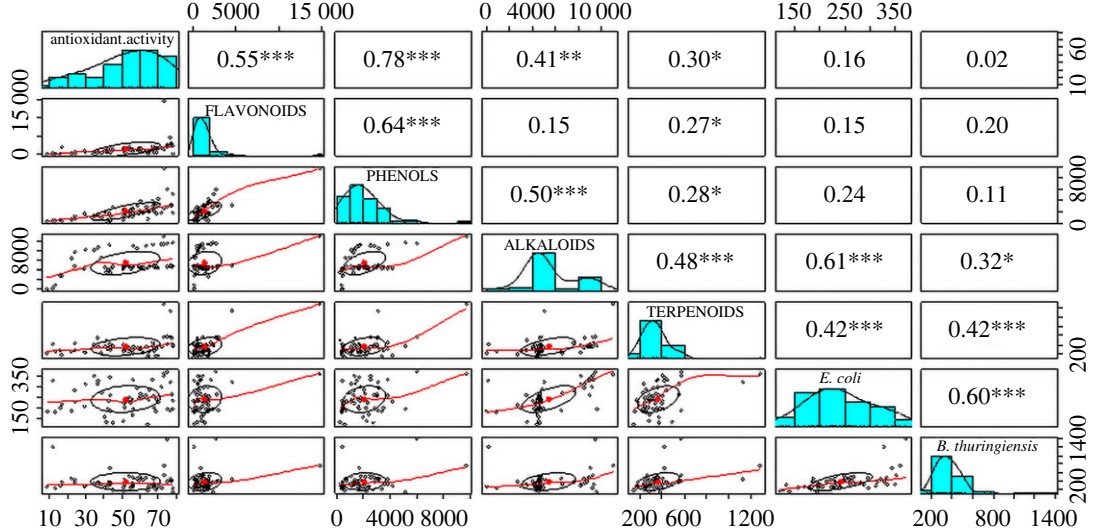

**Figure 4.** Relationship of phytochemical composition, biological activity, and climatic and landscape variables with correlation coefficients, with corresponding significance levels (no star = not statistically significant, *0.05, **0.01, ***0.001).

vegetation, which is influenced by the local climatic conditions [30]. The extracting solvent also affects the concentration of the compounds in different extracts, but water–ethanol as a solvent is preferred [16]. The total phenol and flavonoid content agree with the range recorded by Sun *et al.* [31] of 6.7 to 164.2 mg GAE g$^{-1}$ and 4.1 to 282.8 mg RE g$^{-1}$, which also agrees with quantities recorded by Danert *et al.* [32].

Alkaloids are primarily found in plants and are biologically active compounds with at least one N atom [33]. They are generally bicyclic, tricyclic and tetracyclic derivatives of quinolizidine and the major chemicals in commonly ingested foods and beverages such as coffee, tea leaves and cocoa [34]. Of importance is that most of the plant-derived pharmaceutical products against malaria and cancer are alkaloids [35]. The quantification of total alkaloid and terpenoid content in propolis has not been extensively reported, and our results are the first report. Studies have reported these compounds through GC-MS analysis [36]. Terpenoids are the largest class of naturally occurring products with classification based on isoprene units with a regular arrangement of these units; some are irregular [37]. Hemiterpenoids are found in the leaves of many plants, with monoterpenes found as complexes in plant essential oils. Terpenoids are responsible for the characteristic aroma of propolis.

## 4.2. Antioxidant activity

The DPPH is a stable free radical widely used to evaluate the antioxidant activity of extracts and pure substances. The effect of antioxidants on the DPPH is due to its two mechanisms of action, single electron transfer and hydrogen ion transfer [38]. Propolis displays antioxidant activity due to phytochemicals, which are secondary plant metabolites and are known to act as natural antioxidants. The antioxidant activities of terpenoids, phenols and flavonoids are well documented [39]. The concentration of 250 mg l$^{-1}$ in our study promoted varying antioxidant activity, with the highest being 74%. This is lower than the reported 92.4% [40] for the same propolis concentration using only absolute ethanol as a solvent than our 50% ethanol solvent. The results were close to those posted by Bonamigo *et al.* [41]. The difference is attributed to the solvent used and the geographical location of the propolis collection. Phytochemicals contribute substantially to the antioxidant activity indicated by positive correlations.

## 4.3. Antimicrobial activity

Propolis demonstrated broad-spectrum antibacterial activity against *E. coli* and *B. thuringiensis,* which are Gram-negative and Gram-positive bacteria, respectively. This is in accordance with other studies that showed a similar activity of propolis [39]. Phytochemicals present in propolis are responsible for its antibacterial activity. The GC-MS analysis identified compounds that are known to exhibit antibacterial activity, such as amyrin [40], ferruginol [34] and totarol [28]. The variation in

**Table 3.** Compounds present in propolis identified by GC-MS analysis. Values show the relative abundance ($* 10^6$). Only compounds with similarity score greater than or equal to 90% were considered.

| compound | class | retention time (min) | hot and dry | | hot and wet | | cold and wet | | biological activity |
| --- | --- | --- | --- | --- | --- | --- | --- | --- | --- |
| | | | South Kitui | Marigat | Gede | Kakamega | Murang'a | Taita Hills | |
| octadecahydro-2H-picen-3-one | triterpenoid | 40.1 | 216.0 | 211.0 | 81.0 | 57.0 | 3.0 | 19.0 | antioxidant |
| | triterpenoid | 31.1 | — | — | 10.0 | — | — | — | antioxidant |
| sempervirol | diterpenoid | 26.8 | — | — | 9.0 | 10.0 | — | — | antioxidant |
| lup-20(29)-en-3-ol | triterpenoid | 43.1 | — | 29.0 | — | — | — | — | antioxidant |
| rosadiene | diterpenoid | 26.9 | — | 10.0 | — | — | — | — | antibacterial |
| cubitene | diterpenoid | 23.6 | — | 5.0 | — | — | — | — | antibacterial |
| lupenone | triterpenoid | 31.7 | — | 40.0 | 52.0 | 107.0 | 89.0 | 209.0 | antibacterial |
| alpha-amyrin | triterpenoid | 40.7 | 23 | — | — | — | — | — | antibacterial |
| beta-amyrin | triterpenoid | 39.6 | 13.0 | 89.0 | 197.0 | 40.0 | 4.0 | — | antibacterial |
| oleanen-3-yl acetate | triterpenoid | 43.2 | 475.0 | — | — | — | — | 82.0 | antibacterial |
| ursenoic acid | triterpenoid | 43.2 | 4.0 | — | — | — | — | — | antibacterial |
| lanosta-8,24-dien-3-one | triterpenoid | 38.5 | — | 16.0 | 24.0 | — | — | — | antioxidant |
| eicosane | hydrocarbon | 28.4 | 8.0 | 10.1 | 20.0 | 40.0 | 7.0 | 19.0 | antibacterial |
| heneicosane | hydrocarbon | 34.2 | 24.0 | 62.0 | 23.0 | — | — | — | antibacterial |
| tricosane | hydrocarbon | 26.7 | 11.0 | — | — | — | 10.0 | — | antibacterial |
| nonadecane | hydrocarbon | 31.7 | 4.0 | — | — | — | — | — | no reported bioactivity |
| sugiol | diterpenoid | 29.6 | — | — | 25.0 | 7.0 | — | — | antimicrobial and antioxidant |

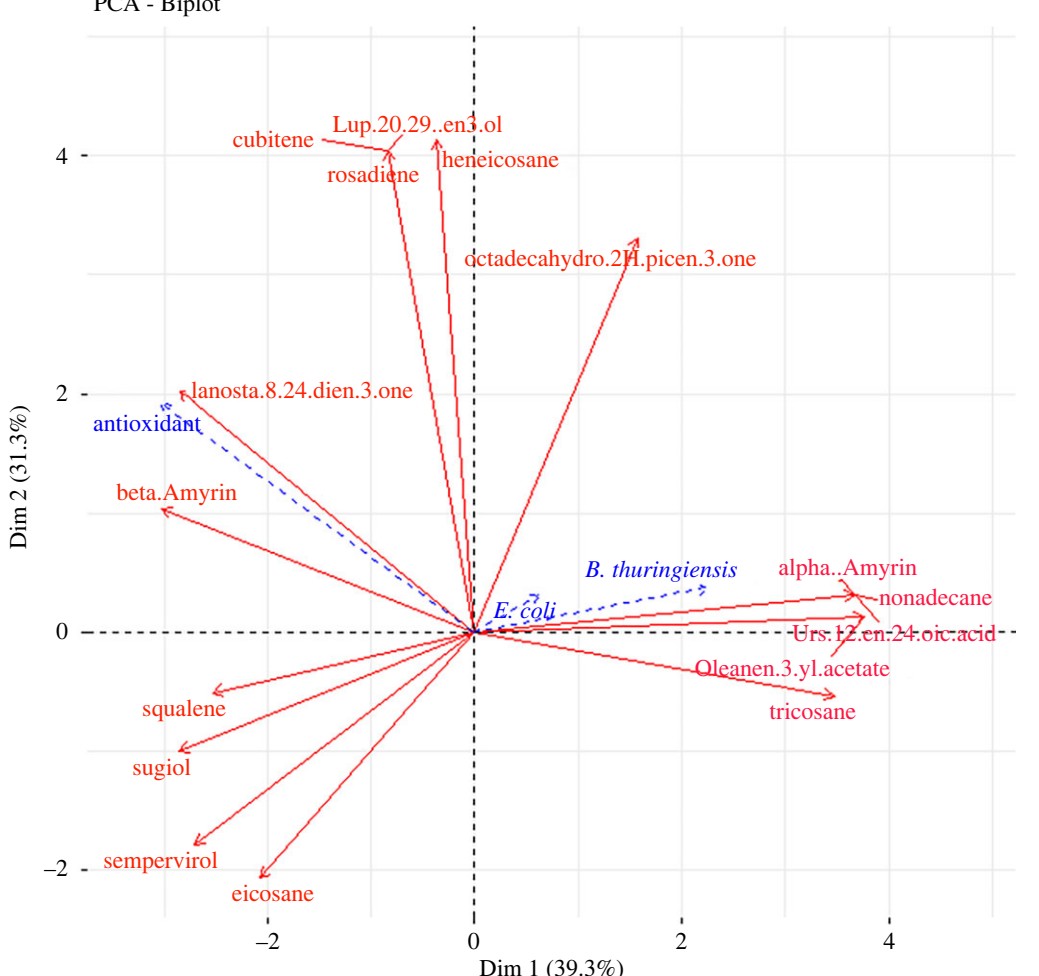

**Figure 5.** Principal components analysis of Kenyan propolis samples based on their bioactivities and non-volatile compounds identified by GC-MS. Total variance explained: 70.6%.

antibacterial activity is due to the presence of these compounds and their concentration in the samples. These results make propolis a potential source of antibacterial compounds for further analysis.

## 4.4. Gas chromatography–mass spectrometry analysis

In this study, the propolis samples showed compounds of variable quantity as reported by others [34]. There was a difference in chemical composition from different climatic conditions because the composition of propolis is directly affected by vegetation [11]. The presence of compounds such as squalene, picenone, sempervirol, amyrin, sugiol and long-chain hydrocarbons such as eicosane have been reported to be components of propolis [42,43]. There are three sources of organic compounds of propolis: compounds derived from the honeybees' metabolism, plant origin and structural materials used to form propolis [42]. These compounds have bioactivities that are attributed to the bioactivities of propolis. The bioactivities for these compounds include amyrin as antibacterial, squalene as an antioxidant, sugiol as antibacterial and antioxidant, and lupenone as an antioxidant [33,44–47].

## 5. Conclusion

The outcome of this study suggests that *Apis mellifera* propolis from Kenya has varying amounts of phytochemicals. This is dependent on the geographical and climatic conditions of different sampling locations. Propolis consisting of plant exudates, buds and resins implies that the vegetation type directly influences its chemical composition. Apart from widely studied phenols and flavonoids, alkaloids were the most abundant phytochemicals with noticeable effects on the bioactivities of

propolis samples. Terpenoids were also abundant in the propolis, and they influenced the antibacterial activities of propolis.

GC-MS analysis can be used to screen bioactive compounds in propolis as it results in different compounds in the propolis samples, and propolis is a rich source of biologically active compounds. The bioactivities of propolis are ascribed to the presence of reported compounds, sugiol, lupenone and squalene, which are phytochemicals identified through GC-MS analysis.

With the exhibited activity of propolis on DPPH free radicals, it is a natural source of antioxidants with high activities in small quantities compared with honey, and this should not be overlooked. Therefore, propolis and its utilization should be encouraged by adopting innovative strategies for its production and harvest.

Data accessibility. The datasets supporting this article have been uploaded as part of the electronic supplementary material [48].

Authors' contributions. T.M.K.: conceptualization, data curation, formal analysis, methodology, writing—original draft; J.L.B.: conceptualization, supervision, writing—review and editing; H.O.M.: data curation, formal analysis, methodology, writing—review and editing; H.M.G.L.: conceptualization, funding acquisition, project administration, resources, supervision, writing—review and editing.

All authors gave final approval for publication and agreed to be held accountable for the work performed therein.

Conflict of interest declaration. We declare we have no competing interests.

Funding. J.L.B. is supported by DELTAS Africa Initiative grant no. DEL-15-011 to THRiVE-2. The DELTAS Africa Initiative is an independent funding scheme of the African Academy of Sciences (AAS)'s Alliance for Accelerating Excellence in Science in Africa (AESA) and supported by the New Partnership for Africa's Development Planning and Coordinating Agency (NEPAD Agency) with funding from the Wellcome Trust grant no. 107742/Z/15/Z and the UK government. The views expressed herein do not necessarily reflect the official opinion of the donors.

Acknowledgements. The authors are indebted to the farmers for providing the propolis samples for the study and the entire environmental theme for their unwavering support. We also acknowledge icipe and the Government of Kenya for all facilitation provided for the accomplishment of this research. We gratefully acknowledge the financial support for this research by the following organizations and agencies: JRS Biodiversity Foundation (grant no. 60610); UK's Foreign, Commonwealth & Development Office (FCDO); the Swedish International Development Cooperation Agency (Sida); the Swiss Agency for Development and Cooperation (SDC); the Federal Democratic Republic of Ethiopia; and the Government of the Republic of Kenya.

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
