## [Peer Review File · Royal Society Open Science]

Review History

RSOS-211214.R0 (Original submission)

Review form: Reviewer 1

Is the manuscript scientifically sound in its present form?

Yes

Are the interpretations and conclusions justified by the results?

No

Is the language acceptable?

Yes

Do you have any ethical concerns with this paper?

No

Have you any concerns about statistical analyses in this paper?

No

Recommendation?

Major revision is needed (please make suggestions in comments)

Comments to the Author(s)

Comments to the author: In this manuscript, the author discussed about Phytochemical composition, antibacterial, and antioxidant properties of *Apis mellifera* propolis from different regions in Kenya. The present study of this kind on natural products is of great help and guidance to the related researchers for further study. The manuscript was organized quite well but still there are some major comments to improve the manuscript.

1. In the discussion parts comparisons of antioxidant potential, antimicrobial potential with previously reported data is recommended.
2. In the discussion parts comparisons of total polyphenolic content, total flavonoid content, total terpenoid content and total alkaloid content is recommended.
3. Discussion part can be elaborated more.
4. Provide the standard equations used for the analysis of total polyphenolic content, total flavonoid content, total terpenoid content.
5. In table number 1 provide how the results are represented (like mg GAE/g and mg QE/g) along with their full form in foot notes.
6. Qualitative analysis of best sample is also recommended in order to find out the complete phytochemical profile.
7. An important question must be answered in the revised manuscript: what component or components are responsible of the antioxidant capacity and antimicrobial potential of the extracts?
8. In order to have the complete profile of polyphenolics HPLC-DAD or LCMS is recommended, if possible, this will provide the more strength to the manuscript.
9. Do go through the following paper for more detail DOI:10.1039/C6RA12038E, DOI: 10.1007/s11694-016-9349-5, DOI: 10.1111/jfbc.12337, <https://doi.org/10.1007/s11694-020-00721-9>, https://doi.org/10.1007/978-981-15-7285-2_1

Review form: Reviewer 2

Is the manuscript scientifically sound in its present form?

No

Are the interpretations and conclusions justified by the results?

No

Is the language acceptable?

No

Do you have any ethical concerns with this paper?

No

Have you any concerns about statistical analyses in this paper?

No

Recommendation?

Major revision is needed (please make suggestions in comments)

Comments to the Author(s)

38 comments

This work listed a short reports on the composition and antibacterial and antioxidant activities of *Apis mellifera* propolis from different regions in Kenya. The topic is interesting, but the results are simple. Based on the given results, I could not know what components are the most of antibacterial or antioxidants. No further components analyses or quantitative results to explain why (the reasons). Therefore, I am not sure that it is suitable for publication in this journal.

Alternatively, author may extract the components and tested their activities, and further deduced the major active components.

Special comments:

- (1) List all data in supporting files including crude data in figures.
- (2) If possible, using HPLC or LC-MS methods to analyze the on-line or off-line activities related components.
- (3) Give clear descriptions of figures. The table may be list in three-line table.

Decision letter (RSOS-211214.R0)

Dear Dr Lattorff:

Title: Phytochemical composition, antibacterial, and antioxidant properties of *Apis mellifera* propolis from different regions in Kenya
Manuscript ID: RSOS-211214

The editor assigned to your manuscript has now received comments from reviewers. We would like you to revise your paper in accordance with the referee and Subject Editor suggestions which can be found below (not including confidential reports to the Editor). Please note this decision does not guarantee eventual acceptance.

Please submit your revised paper before 27-Nov-2021. Please note that the revision deadline will expire at 00.00am on this date. If we do not hear from you within this time then it will be assumed that the paper has been withdrawn. In exceptional circumstances, extensions may be possible if agreed with the Editorial Office in advance. We do not allow multiple rounds of revision so we urge you to make every effort to fully address all of the comments at this stage. If deemed necessary by the Editors, your manuscript will be sent back to one or more of the original reviewers for assessment. If the original reviewers are not available we may invite new reviewers.

Please also include the following statements alongside the other end statements. As we cannot publish your manuscript without these end statements included, if you feel that a given heading is not relevant to your paper, please nevertheless include the heading and explicitly state that it is not relevant to your work.

- Ethics statement

Please clarify whether you received ethical approval from a local ethics committee to carry out your study. If so please include details of this, including the name of the committee that gave consent in a Research Ethics section after your main text. Please also clarify whether you received informed consent for the participants to participate in the study and state this in your Research Ethics section.

OR

Please clarify whether you obtained the necessary licences and approvals from your institutional animal ethics committee before conducting your research. Please provide details of these licences and approvals in an Animal Ethics section after your main text.

OR

Please clarify whether you obtained the appropriate permissions and licences to conduct the fieldwork detailed in your study. Please provide details of these in your methods section.

- Data accessibility

It is a condition of publication that you make available the data and research materials supporting the results in the article. Datasets should be deposited in an appropriate publicly available repository and details of the associated accession number, link or DOI to the datasets must be included in the Data Accessibility section of the article (<https://royalsocietypublishing.org/rsos/for-authors#question17>). Reference(s) to datasets should also be included in the reference list of the article with DOIs (where available).

Please include a Data Availability section after your main text stating where supporting data are available from, or where they will be made available should your article be accepted for publication.

<http://datadryad.org/submit?journalID=RSOS&manu=RSOS-211214>

- Competing interests

Please include a Competing Interests section after your main text declaring any financial or non-financial competing interests. If you have no competing interests please state 'I/we have no competing interests.'

- Authors' contributions

Please include an Authors' Contributions section at the end of your main text detailing the contribution of each author. All authors should have read and approved the manuscript before submission and this should be stated in the Authors' Contributions section.

The list of Authors should meet all of the following criteria; 1) substantial contributions to conception and design, or acquisition of data, or analysis and interpretation of data; 2) drafting the article or revising it critically for important intellectual content; and 3) final approval of the version to be published.

- Acknowledgements

- Funding statement

Please include a funding section after your main text which lists the source of funding for each author.

Yours sincerely,
Dr Ellis Wilde
Publishing Editor, Journals

On behalf of the Subject Editor Professor Anthony Stace and the Associate Editor Dr Nadia Martinez Villegas.

RSC Associate Editor

Comments to the Author:

The research presented in this draft is original and might be of interest to RSOS audience. However, questions remain on the components that are responsible for the antioxidant capacity and antimicrobial potential of the extracts. Additional HPLC analyses are recommended to strengthen the manuscript.

RSC Subject Editor

Comments to the Author:

(There are no comments.)

Reviewers' Comments to Author:

Reviewer: 1

Comments to the Author(s)

Comments to the author: In this manuscript, the author discussed about Phytochemical composition, antibacterial, and antioxidant properties of *Apis mellifera* propolis from different regions in Kenya. The present study of this kind on natural products is of great help and guidance to the related researchers for further study. The manuscript was organized quite well but still there are some major comments to improve the manuscript.

1. In the discussion parts comparisons of antioxidant potential, antimicrobial potential with previously reported data is recommended.
2. In the discussion parts comparisons of total polyphenolic content, total flavonoid content, total terpenoid content and total alkaloid content is recommended.
3. Discussion part can be elaborated more.
4. Provide the standard equations used for the analysis of total polyphenolic content, total flavonoid content, total terpenoid content.
5. In table number 1 provide how the results are represented (like mg GAE/g and mg QE/g) along with their full form in foot notes.
6. Qualitative analysis of best sample is also recommended in order to find out the complete phytochemical profile.
7. An important question must be answered in the revised manuscript: what component or components are responsible of the antioxidant capacity and antimicrobial potential of the extracts?
8. In order to have the complete profile of polyphenolics HPLC-DAD or LCMS is recommended, if possible, this will provide the more strength to the manuscript.
9. Do go through the following paper for more detail DOI:10.1039/C6RA12038E, DOI: 10.1007/s11694-016-9349-5, DOI: 10.1111/jfbc.12337, <https://doi.org/10.1007/s11694-020-00721-9>, https://doi.org/10.1007/978-981-15-7285-2_1

Reviewer: 2

Comments to the Author(s)

38 comments

This work listed a short reports on the composition and antibacterial and antioxidant activities of *Apis mellifera* propolis from different regions in Kenya. The topic is interesting, but the results are simple. Based on the given results, I could not know what components are the most of antibacterial or antioxidants. No further components analyses or quantitative results to explain why (the reasons). Therefore, I am not sure that it is suitable for publication in this journal.

Alternatively, author may extract the components and tested their activities, and further deduced the major active components.

Special comments:

- (1) List all data in supporting files including crude data in figures.
- (2) If possible, using HPLC or LC-MS methods to analyze the on-line or off-line activities related components.
- (3) Give clear descriptions of figures. The table may be list in three-line table.

Author's Response to Decision Letter for (RSOS-211214.R0)

See Appendix A.

RSOS-211214.R1 (Revision)

Review form: Reviewer 1

Is the manuscript scientifically sound in its present form?

Yes

Are the interpretations and conclusions justified by the results?

Yes

Is the language acceptable?

Yes

Do you have any ethical concerns with this paper?

No

Have you any concerns about statistical analyses in this paper?

No

Recommendation?

Accept as is

Comments to the Author(s)

The authors have addressed all the comments and have also done the required changes in the manuscript. Now, it is acceptable in present form.

Review form: Reviewer 2

Is the manuscript scientifically sound in its present form?

No

Are the interpretations and conclusions justified by the results?

Yes

Is the language acceptable?

Yes

Do you have any ethical concerns with this paper?

No

Have you any concerns about statistical analyses in this paper?

No

Recommendation?

Accept with minor revision (please list in comments)

Comments to the Author(s)

Several concerns have been addressed. But the data of antioxidant and antibacterial activities of samples should list related results of a positive compound as control at least before publication.

Decision letter (RSOS-211214.R1)

Dear Dr Lattorff:

Title: Phytochemical composition and bio-functional properties of *Apis mellifera* propolis from Kenya
Manuscript ID: RSOS-211214.R1

Thank you for submitting the above manuscript to Royal Society Open Science. On behalf of the Editors and the Royal Society of Chemistry, I am pleased to inform you that your manuscript will be accepted for publication in Royal Society Open Science subject to minor revision in accordance with the referee suggestions. Please find the reviewers' comments at the end of this email.

The reviewers and handling editors have recommended publication, but also suggest some minor revisions to your manuscript. Therefore, I invite you to respond to the comments and revise your manuscript.

Please also include the following statements alongside the other end statements. As we cannot publish your manuscript without these end statements included, if you feel that a given heading is not relevant to your paper, please nevertheless include the heading and explicitly state that it is not relevant to your work. We have included a screenshot example of the end statements for reference.

- Ethics statement

Please clarify whether you received ethical approval from a local ethics committee to carry out your study. If so please include details of this, including the name of the committee that gave consent in a Research Ethics section after your main text. Please also clarify whether you received informed consent for the participants to participate in the study and state this in your Research Ethics section.

OR

Please clarify whether you obtained the necessary licences and approvals from your institutional animal ethics committee before conducting your research. Please provide details of these licences and approvals in an Animal Ethics section after your main text.

OR

Please clarify whether you obtained the appropriate permissions and licences to conduct the fieldwork detailed in your study. Please provide details of these in your methods section.

- Data accessibility

It is a condition of publication that you make available the data and research materials supporting the results in the article. Datasets should be deposited in an appropriate publicly available repository and details of the associated accession number, link or DOI to the datasets must be included in the Data Accessibility section of the article (<https://royalsocietypublishing.org/rsos/for-authors#question17>). Reference(s) to datasets should also be included in the reference list of the article with DOIs (where available).

Please include a Data Availability section after your main text stating where supporting data are available from, or where they will be made available should your article be accepted for publication.

If you wish to submit your supporting data or code to Dryad (<http://datadryad.org/>), or modify your current submission to dryad, please use the following link:
<http://datadryad.org/submit?journalID=RSOS&manu=RSOS-211214.R1>

- **Competing interests**

Please include a Competing Interests section after your main text declaring any financial or non-financial competing interests. If you have no competing interests please state 'I/we have no competing interests.'

- **Authors' contributions**

Please include an Authors' Contributions section at the end of your main text detailing the contribution of each author. All authors should have read and approved the manuscript before submission and this should be stated in the Authors' Contributions section.

The list of Authors should meet all of the following criteria; 1) substantial contributions to conception and design, or acquisition of data, or analysis and interpretation of data; 2) drafting the article or revising it critically for important intellectual content; and 3) final approval of the version to be published.

- **Acknowledgements**

- **Funding statement**

Please include a funding section after your main text which lists the source of funding for each author.

Because the schedule for publication is very tight, it is a condition of publication that you submit the revised version of your manuscript before 03-Mar-2022. Please note that the revision deadline will expire at 00.00am on this date. If you do not think you will be able to meet this date please let me know immediately.

When submitting your revised manuscript, you will be able to respond to the comments made by the referees and upload a file "Response to Referees" in "Section 6 - File Upload". You can use this to document any changes you make to the original manuscript. In order to expedite the

processing of the revised manuscript, please be as specific as possible in your response to the referees.

Kind regards,
Kate Jones
Publishing Editor, Journals

On behalf of the Subject Editor Professor Anthony Stace and the Associate Editor Dr Nadia Martinez Villegas.

RSC Associate Editor

Comments to the Author:

Although the authors attended most of the comments satisfactorily, one of the reviewers is still pointing out the need for data of antioxidant and antibacterial activities. Please add this information to your manuscript and be aware that we previously asked you to make the effort to fully address all the comments, as we do not allow multiple rounds of revisions. We look forward

to receiving the revised version of your manuscript exhaustively analyzing all relevant scientific and methodological issues.

RSC Subject Editor

Comments to the Author:

(There are no comments.)

Reviewer comments to Author:

Reviewer: 2

Comments to the Author(s)

Several concerns have been addressed. But the data of antioxidant and antibacterial activities of samples should list related results of a positive compound as control at least before publication.

Reviewer: 1

Comments to the Author(s)

The authors have addressed all the comments and have also done the required changes in the manuscript. Now, it is acceptable in present form.

Author's Response to Decision Letter for (RSOS-211214.R1)

See Appendix B.

RSOS-211214.R2

Review form: Reviewer 1

Is the manuscript scientifically sound in its present form?

Yes

Are the interpretations and conclusions justified by the results?

Yes

Is the language acceptable?

Yes

Do you have any ethical concerns with this paper?

No

Have you any concerns about statistical analyses in this paper?

No

Recommendation?

Major revision is needed (please make suggestions in comments)

Comments to the Author(s)

The manuscript was organized quite well but still there are some major comments to improve the manuscript.

1. The manuscript is well written, although a grammar review is recommended.
2. Table 3 octadecahydro-2H-picen-3-one and squalene are not a flavonoid, these are triterpene, Lup-20(29)-en-3-ol is also triterpene, Sempervirol is diterpenoid. Kindly again review the table 3 and correct it accordingly.

Decision letter (RSOS-211214.R2)

Dear Dr Lattorff:

Title: Phytochemical composition and bio-functional properties of *Apis mellifera* propolis from Kenya

Manuscript ID: RSOS-211214.R2

Thank you for submitting the above manuscript to Royal Society Open Science. On behalf of the Editors and the Royal Society of Chemistry, I am pleased to inform you that your manuscript will be accepted for publication in Royal Society Open Science subject to minor revision in accordance with the referee suggestions. Please find the reviewers' comments at the end of this email.

The reviewers and handling editors have recommended publication, but also suggest some minor revisions to your manuscript. Therefore, I invite you to respond to the comments and revise your manuscript.

Please also include the following statements alongside the other end statements. As we cannot publish your manuscript without these end statements included, if you feel that a given heading is not relevant to your paper, please nevertheless include the heading and explicitly state that it is not relevant to your work. We have included a screenshot example of the end statements for reference.

- Ethics statement

Please clarify whether you received ethical approval from a local ethics committee to carry out your study. If so please include details of this, including the name of the committee that gave consent in a Research Ethics section after your main text. Please also clarify whether you received informed consent for the participants to participate in the study and state this in your Research Ethics section.

OR

Please clarify whether you obtained the necessary licences and approvals from your institutional animal ethics committee before conducting your research. Please provide details of these licences and approvals in an Animal Ethics section after your main text.

OR

Please clarify whether you obtained the appropriate permissions and licences to conduct the fieldwork detailed in your study. Please provide details of these in your methods section.

- Data accessibility

It is a condition of publication that you make available the data and research materials supporting the results in the article. Datasets should be deposited in an appropriate publicly available repository and details of the associated accession number, link or DOI to the datasets must be included in the Data Accessibility section of the article (<https://royalsocietypublishing.org/rsos/for-authors#question17>). Reference(s) to datasets should also be included in the reference list of the article with DOIs (where available).

Please include a Data Availability section after your main text stating where supporting data are available from, or where they will be made available should your article be accepted for publication.

If you wish to submit your supporting data or code to Dryad (<http://datadryad.org/>), or modify your current submission to dryad, please use the following link:
<http://datadryad.org/submit?journalID=RSOS&manu=RSOS-211214.R2>

- Competing interests

Please include a Competing Interests section after your main text declaring any financial or non-financial competing interests. If you have no competing interests please state 'I/we have no competing interests.'

- Authors' contributions

Please include an Authors' Contributions section at the end of your main text detailing the contribution of each author. All authors should have read and approved the manuscript before submission and this should be stated in the Authors' Contributions section.

The list of Authors should meet all of the following criteria; 1) substantial contributions to conception and design, or acquisition of data, or analysis and interpretation of data; 2) drafting the article or revising it critically for important intellectual content; and 3) final approval of the version to be published.

- Acknowledgements

- Funding statement

Please include a funding section after your main text which lists the source of funding for each author.

Because the schedule for publication is very tight, it is a condition of publication that you submit the revised version of your manuscript before 06-May-2022. Please note that the revision deadline will expire at 00.00am on this date. If you do not think you will be able to meet this date please let me know immediately.

Kind regards,
Zita Zachariah, PhD
Assistant Editor, Journals

On behalf of the Subject Editor Professor Anthony Stace and the Associate Editor Dr Nadia Martinez Villegas.

RSC Associate Editor: 1

Comments to the Author:

The authors of this manuscript replied satisfactorily to most of the comments. However, a few more corrections need to be addressed before publication. Please read carefully each of the comments from the reviewers and make sure that you address each of them satisfactorily.

RSC Associate Editor: 2

Comments to the Author:

(There are no comments.)

Reviewer comments to Author:

Reviewer: 1

Comments to the Author(s)

The manuscript was organized quite well but still there are some major comments to improve the manuscript.

1. The manuscript is well written, although a grammar review is recommended.
2. Table 3 octadecahydro-2H-picen-3-one and squalene are not a flavonoid, these are triterpene, Lup-20(29)-en-3-ol is also triterpene, Sempervirol is diterpenoid. Kindly again review the table 3 and correct it accordingly.

Author's Response to Decision Letter for (RSOS-211214.R2)

See Appendix C.

Decision letter (RSOS-211214.R3)

Dear Dr Lattorff:

Title: Phytochemical composition and bio-functional properties of *Apis mellifera* propolis from Kenya

Manuscript ID: RSOS-211214.R3

It is a pleasure to accept your manuscript in its current form for publication in Royal Society Open Science. The chemistry content of Royal Society Open Science is published in collaboration with the Royal Society of Chemistry.

Where applicable, the comments of the reviewer(s) who reviewed your manuscript are included at the end of this email.

If you have not already done so, please ensure that you send to the editorial office (openscience@royalsociety.org) an editable version of your accepted manuscript, and individual

files for each figure and table included in your manuscript. You can send these in a zip folder if more convenient. Failure to provide these files may delay the processing of your proof.

Please remember to make any data sets or code libraries 'live' prior to publication, and update any links as needed when you receive a proof to check - for instance, from a private 'for review' URL to a publicly accessible 'for publication' URL. It is also good practice to add data sets, code and other digital materials to your reference list.

Royal Society Open Science is a fully open access journal. A payment may be due before your article is published. Our partner Copyright Clearance Centre will contact the corresponding author about your open access options (if you have any queries regarding fees, please see <https://royalsocietypublishing.org/rsos/charges> or contact authorfees@royalsociety.org).

Yours sincerely,
Raffaele Egizio
Assistant Editor, Journals

On behalf of the Subject Editor Professor Anthony Stace and the Associate Editor Dr Nadia Martinez Villegas.

RSC Associate Editor
Comments to the Author:
The paper's authors addressed the reviewer's comments satisfactorily. The article can now be accepted.

Reviewer(s)' Comments to Author:

Appendix A

Response to Referees

Bold letter = our responses

Reviewer: 1

1. In the discussion parts comparisons of antioxidant potential, antimicrobial potential with Previously reported data is recommended.

Thanks for the above comment which will improve the quality of our work. We have addressed it by having the discussion in two paragraphs. We have added references in the discussion to address the comparisons with other reported data. This is in lines 348-360 for antioxidant activity and lines 362-370 for antimicrobial activity. These sections are accompanied with eight references.

2. In the discussion parts comparisons of total polyphenolic content, total flavonoid content, total terpenoid content and total alkaloid content is recommended.

Thanks for the comment, and we have addressed by having a section with two paragraphs. First paragraph addresses total phenolic content and total flavonoid content (lines 329-335) and the second paragraph addressing total alkaloid content and total flavonoid content (lines 336-347). There is comparison of with reported data accompanied with references.

3. Discussion part can be elaborated more.

We are grateful for the above comment. We have addressed it by having four sections in the discussion part: Quantification of phytochemicals, Antioxidant activity, Anti-microbial activity and GC-MS analysis. All these sections have references on previously reported data

4. Provide the standard equations used for the analysis of total polyphenolic content, total flavonoid content, total terpenoid content.

Thanks for the comment we noted it and addressed it by adding all the equations for the phytochemicals. Total phenol content line 143 ($y = 0.0073x + 0.0233$, $R^2 = 0.9992$), total flavonoid content line 133 ($y = 0.0006x + 0.0028$, $R^2 = 0.9981$), total alkaloid content line 155 ($y = 1.866x + 0.2332$, $R^2 = 0.9844$) and total terpenoid content line 167 ($y = 0.0009x - 0.0158$, $R^2 = 0.9914$)

5. In table number 1 provide hoe the results are represented (like mg GAE/g and mg QE/g) along with their full form in foot notes.

We are grateful for the observation, and we have addressed this by providing the full forms in the footnotes, lines 298-299 (QE - Quercetin Equivalent, GAE - Gallic Acid Equivalent, LE - Linalool Equivalent, CE - Colchicine Equivalent). Our results were for 100g as we had indicated in the table title line 294 (Table 2. Phytochemical components of studied propolis in mg/100 g propolis).

6. Qualitative analysis of best sample is also recommended in order to find out the complete phytochemical profile.

We are grateful and really do appreciate the comment and we did qualitative analysis of six samples using GCMS and we obtained the profiles, and this data is in our revised manuscript.

7. An important question must be answered in the revised manuscript: what component or components are responsible of the antioxidant capacity and antimicrobial potential of the extracts?

Thanks again for the comment which will make our manuscript more informative. We have addressed this by doing GCMS analysis and inferring to the compounds present and analysing their bioactivities in line with either antibacterial or antioxidant activity (Table 3. Compounds present in propolis identified by GC-MS analysis. Values show the relative abundance (* 10⁶). Only compounds with similarity score ≥ 90% were considered). Compounds from the profile have either of these bioactivities and hence are the reason why propolis exhibits the antioxidant or antimicrobial potential. The above question is answered in that we can tell the compounds responsible for the difference in either antimicrobial or antioxidant activity from Table 3 and figure 5.

8. In order to have the complete profile of polyphenolics HPLC-DAD or LCMS is recommended, if possible, this will provide the more strength to the manuscript.

We are grateful on this comment which is a great observation and very helpful. We addressed this in a different way but with the same output. We did GCMS analysis by liquid-liquid extraction where dichloromethane is used as a solvent (line 172-188). This resulted to different chemical profiles based on presence or absence of it and the varying abundance (Table 3).

9. Do go through the following paper for more detail DOI:10.1039/C6RA12038E, DOI: 10.1007/s11694-016-9349-5, DOI: 10.1111/jfbc.12337, <https://doi.org/10.1007/s11694-020-00721-9>, https://doi.org/10.1007/978-981-15-7285-2_1

We went through the above papers and have been of great help in addressing the comments and we are grateful for this gesture.

Reviewer: 2

Comments to the Author(s)

This work listed a short reports on the composition and antibacterial and antioxidant activities of *Apis mellifera* propolis from different regions in Kenya. The topic is interesting, but the results are

simple. Based on the given results, I could not know what components are the most of antibacterial or antioxidants. No further components analyses or quantitative results to explain why (the reasons). Therefore, I am not sure that it is suitable for publication in this journal.

Alternatively, author may extract the components and tested their activities, and further deduced the major active components.

Thanks for the above comment it has helped us to refine our revised manuscript. In our revised manuscript we have done qualitative analysis of selected samples using liquid-liquid extraction GCMS analysis and presented this in table 3 and figure 5.

Special comments:

(1) List all data in supporting files including crude data in figures.

We do appreciate the comment and we have addressed it.

(2) If possible, using HPLC or LC-MS methods to analyze the on-line or off-line activities related components.

Thanks for the above comment and suggestion. We addressed this in a different way but with the same output. We did GCMS analysis by liquid-liquid extraction where dichloromethane is used as a solvent (line 172-188). This resulted to different chemical profiles (table 3 and figure 5).

(3) Give clear descriptions of figures. The table may be list in three-line table.

We are grateful for the comment. We have addressed this by restructuring and putting additional information on apiaries in table 1, and we have also added on the footnotes in table 2 (lines 298-299 (QE - Quercetin Equivalent, GAE - Gallic Acid Equivalent, LE - Linalool Equivalent, CE - Colchicine Equivalent). Our results were for 100g as we had indicated in the table title line 294).

Appendix B

Response to Referees

 = Responses

Reviewer: 1

1. The authors have addressed all the comments and have also done the required changes in the manuscript. Now, it is acceptable in present form.

We are really grateful for the above comment and we also acknowledge your priceless contribution and support to this work. Thanks.

Reviewer: 2

Comments to the Author(s)

Several concerns have been addressed. But the data of antioxidant and antibacterial activities of samples should list related results of a positive compound as control at least before publication.

Thanks for the above comment it is of great help and a very important observation which will help us refine our revised manuscript. In our revised manuscript we have addressed the above comments in as follows:

Positive control for antioxidant activity

We used quercetin as positive control for the antioxidant activity to come up with a standard curve (10-100 µg/mL) in (line 221-22) which was used to correlate the percentage inhibition of our results (line 271-274) (Table S2, Table S3 and Figure S1).

Positive control for antibacterial activity

For the antibacterial activity, the positive control was streptomycin which is an antibiotic was used (line 208-209) and the results correlated with the positive control (line 283-284) (Table S4).

We have provided the data for all positive controls in the supplementary material.

Appendix C

Response to Referees

 = Responses

RSC Associate Editor: 2

Comments to the Author:

(There are no comments.)

Reviewer comments to Author:

Reviewer: 1

Comments to the Author(s)

The manuscript was organized quite well but still there are some major comments to improve the manuscript.

1. The manuscript is well written, although a grammar review is recommended.

Thanks a lot, we have reviewed and changed accordingly when we found grammar issues.

2. Table 3 octadecahydro-2H-picen-3-one and squalene are not a flavonoid, these are triterpene, Lup-20(29)-en-3-ol is also triterpene, Sempervirol is diterpenoid. Kindly again review the table 3 and correct it accordingly.

We replaced Table 3 and inserted the suggested changes.